# Far-UVC light (222 nm) efficiently inactivates clinically significant antibiotic-resistant bacteria on diverse material surfaces

Jhen-Rong Huang,[1] Tsai-Wen Yang,[1] Ya-I Hsiao,[1] Hui-Min Fan,[1] Han-Yueh Kuo,[2] Kuo-Hsiang Hung,[1] Po-Yen Chen,[3] Ching-Ting Tan,[4,5] Pei-Lan Shao[1,6]

**ABSTRACT** In recent years, there has been a gradual increase in the prevalence of drug-resistant bacteria, primarily attributed to the widespread use of antibiotics. This has resulted in heightened mortality rates, morbidity, and exorbitant healthcare costs associated with antibiotic-resistant bacterial infections. In order to mitigate the spread of antibiotic-resistant bacteria, environmental disinfection plays a crucial role. Ultraviolet radiation C (UVC) light disinfection has emerged as a potent technique to limit the transmission of nosocomial pathogens and prevent healthcare-associated infections. Different types of high-touch surfaces were used. A serial disinfected experiment with different 222 nm UVC dosages was conducted on clinically isolated antibiotic-resistant bacteria, including methicillin-resistant *Staphylococcus aureus* (MRSA), vancomycin-resistant *Enterococcus* species (VRE), carbapenem-resistant *Escherichia coli* (CREC), carbapenem-resistant *Klebsiella pneumonia* (CRKP), carbapenem-resistant *Acinetobacter baumannii* (CRAB), and carbapenem-resistant *Pseudomonas aeruginosa* (CRPA) on different material surfaces. The bactericidal efficacy was evaluated by The Clinical & Laboratory Standards Institute (CLSI) guidelines. 222 nm UVC irradiation had a potent bactericidal efficacy on clinical antibiotic-resistant bacteria on different high-touch surfaces that are commonly found in the environment and healthcare facilities. 222 nm UVC irradiation time was tested from 10 s to 1 h. Different surfaces affect the efficiency of 222 nm UVC. The more adsorptive a material is, the higher the dosage of 222 nm UVC irradiation energy is required for effective disinfection. The use of 222 nm UVC lamps for disinfection on different materials has been shown to be a useful method. However, it is crucial to pay attention to the energy required for effective sterilization.

**IMPORTANCE** This study is crucial, providing compelling evidence on Far-ultraviolet radiation C (Far-UVC) light's efficacy against clinically significant antibiotic-resistant bacteria—a pressing issue in microbiology and infection control. Our research employs antibiotic-resistant strains from clinically isolated bacteria, emphasizing real-world relevance. Simultaneously, we assess Far-UVC light (222 nm) across diverse material surfaces commonly found in clinical settings. This dual approach ensures practical applicability and broad relevance. Our comprehensive setup and rigorous methodologies unequivocally demonstrate Far-UVC light's potency in combating antibiotic-resistant bacteria. Since 222 nm far-UVC has a disinfection capability and is harmless to mammalian cells, this dual effectiveness positions Far-UVC as a secure tool for infection control, with potential applications in healthcare settings, mitigating antibiotic-resistant bacteria spread, and reducing healthcare-associated infections.

**KEYWORDS** 222 nm UVC light, antibiotic-resistant bacteria, Gram-positive bacteria, Gram-negative bacteria, high-touch surfaces

Address correspondence to Pei-Lan Shao, G41051@hch.gov.tw.

Jhen-Rong Huang and Tsai-Wen Yang contributed equally to this article. The author order was determined alphabetically.

The authors declare no conflict of interest.

In recent years, the widespread use of antibiotics has led to the evolution of bacteria, resulting in resistance to nearly all antibiotics currently available in clinical practice. Antibiotic resistance has developed rapidly over the past few decades and has become one of the greatest public health threats of the 21st century (1). Antibiotic-resistant bacteria increase mortality, morbidity, and excessive healthcare costs (2). The health-care-associated pathogens including methicillin-resistant *Staphylococcus aureus* (MRSA), vancomycin-resistant *Enterococcus* species (VRE), carbapenem-resistant gram-negative bacilli, such as *Escherichia coli*, *Klebsiella pneumonia*, *Pseudomonas aeruginosa*, and *Acinetobacter baumannii* make an important contribution to hospital infection (3–5). The phenomenon has caused more difficulties in clinical treatment for bacteria-infection patients.

Previous studies have confirmed that contaminated environmental surfaces are important sources of spreading healthcare-associated infections (HAIs) (2). In order to reduce antibiotic-resistant bacteria infections, the disinfection of the environment is very important. However, traditional cleaning (e.g., detergent or alcohol wipes) of high-touch surfaces does not always remove pathogens (5). Several pathogens are highly resistant to alcohol-based disinfectants (6). Otherwise, long-term treatment with alcohol possibly deteriorates the materials. Traditional manual cleaning is restricted by the components of the equipment.

Fortunately, ultraviolet radiation C (UVC) light disinfection has been used to limit the transmission of nosocomial pathogens and prevent HAIs in recent years (7). Most UV disinfection uses germicidal lamps emitting UVC around 254 nm. The mechanism of 254 nm UVC light is mainly related to damage DNA or RNA, which often leads to pyrimidine dimerization, causing the death of pathogens (8). However, 254 nm UVC light is known for its hazardous nature to mammalian cells and could lead to dermatitis and skin cancer (9). Previous studies showed that 222 nm UVC light is harmless to mammalian skin (10). This is due to the low penetration of Far-UVC light in human cells of skin or eyes, being absorbed by the stratum corneum layer before reaching the nuclei of the epidermal cells (11). UVC has emerged as an effective strategy for microbial control in indoor public spaces to minimize the risk of pathogens' contamination and propagation (12). UVC blocks airborne and droplet-transmission respiratory tract viruses through space disinfection. UVC can efficiently disinfect viruses present in aerosol, such as coronaviruses and influenza (13, 14). Previous studies have shown that UVC radiation with a wavelength of 222 nm can effectively inactivate a wide range of microbial pathogens. Notably, 222 nm UVC demonstrates similar bactericidal efficacy against bacterial vegetative cells, yeasts, and viruses when compared to 254 nm UVC (15). However, its ability to sterilize antibiotic-resistant bacteria transmitted through surface contact has not been definitively established. Currently, there is a lack of sufficient research regarding the sterilization effectiveness of 222 nm UVC light on clinically isolated antibiotic-resistant bacteria.

Because of the lack of information, we were interested in the sterilized efficiency of 222 nm UVC light against antibiotic-resistant bacteria on different high-touch surfaces, which are commonly found in the environment and healthcare facilities.

## MATERIALS AND METHODS

### UVC light source

The UVC disinfection device used was Delta U+ Disinfection Device Care 222 Series (Delta Electronics, Inc., Taiwan), which is capable of emitting 222 nm wavelength ultraviolet rays. The UVC lamp provides 0.92 mW/cm$^2$ irradiation at 10 cm from the emission window (Table 1).

### High-touch surfaces

The high-touch surfaces that are commonly used in clinical environments and health-care facilities were divided into three types in our study. Initially, the materials with

**TABLE 1** UVC exposure energy of Delta U+ Disinfection Device Care 222 Series (Delta Electronics, Inc., Taiwan) at 10 cm from the emission window

| UVC exposure | | | | | |
|---|---|---|---|---|---|
| Time (s) | 0 | 10 | 20 | 30 | 40 | 50 |
| Energy (mJ/cm$^2$) | 0.0 | 9.2 | 18.3 | 27.5 | 36.6 | 45.8 |
| Time | 60 s | 90 s | 120 s | 5 min | 10 min | 15 min |
| Energy (mJ/cm$^2$) | 55.2 | 82.8 | 110.4 | 276.0 | 552.0 | 828.0 |
| Time | 30 min | 1 h | | | | |
| Energy (mJ/cm$^2$) | 1,656.0 | 3,312.0 | | | | |

smooth and watertight surfaces were represented by a melamine board. In addition, the materials with microporous structures and weak absorption were represented by silicone rubber. Furthermore, the materials with uneven surfaces and strong absorption were represented by wood veneer. These three types of materials cover most textures encountered in hospitals.

## Bacterial strains and culture conditions

Clinical isolates of MRSA, VRE, carbapenem-resistant *Escherichia coli* (CREC), carbapenem-resistant *Klebsiella pneumonia* (CRKP), carbapenem-resistant *Acinetobacter baumannii* (CRAB), and carbapenem-resistant *Pseudomonas aeruginosa* (CRPA) were randomly selected from three strains from different specimens and were used for this *in vitro* study. The information about the bacterial isolates is shown in Table 2. This study is approved by IRB. The ethical protocol number is NTUH-REC No. 202301215W. All strains were stored at −80°C in a CMP GermBank storage tube. Working strains were stored at 4°C and propagated before being used in UVC disinfection. The six species of bacteria mentioned above were grown on 5% sheep blood in Tryptic Soy Agar (TSA) plate at 37°C in a 5% $CO_2$ incubator for 16 h.

## Quantification of colonies

The bacteria were collected by eSwab (Copan's Liquid Amies Elution Swab) and were plated on the TSA plate containing 5% sheep blood and incubated at 37°C for 16 h. Colony-forming units (CFUs) were measured by serial dilution.

## Disinfection by UVC light on different high-touch surfaces

Each isolated strain was suspended in 0.45% sodium chloride solution, which is used for drug susceptibility testing of clinical bacteria. Among that, $1.5 \times 10^7$ CFU/mL of bacteria were spread on melamine board, silicone, and wood veneer and then irradiated with 222 nm UVC light. The materials were all pretreated with 75% alcohol and 256 nm-UV light for 30 min before the experiments. The UVC energy dosages were gradually extended from 9.2 to 110.4 mJ/cm$^2$, corresponding to exposure under UVC light from 10 to 120 s. The prolonged exposure energy was 276.0, 552.0, 828.0, 1656.0, and 3312.0 mJ/cm$^2$, corresponding to 5, 10, 15, 30 min, and 1 h. eSwab collected a suspension of 2.5 cm$^2$ area by rolling it back and forth 10 times and inoculating it onto a TSA agar plate.

## Statistical analysis

Each antibiotic-resistant bacteria group contained three clinical isolates. The replicate determinations were performed two times in each independent assay, and independent assays were performed three times for each isolate. The statistical analyses were performed by using one-way ANOVA.

**TABLE 2** The information of three strains of each antibiotic-resistant bacteria group

| Isolate no. | 1 | 2 | 3 |
|---|---|---|---|
| MRSA | | | |
| Specimens | Respiratory tract | Blood | Wound |
| Oxacillin | Resistant | Resistant | Resistant |
| Sulfamethoxazole-trimethoprim | Sensitive | Sensitive | Sensitive |
| VRE | | | |
| Specimens | Urine | Blood | Wound |
| Vancomycin | Resistant | Resistant | Resistant |
| Linezolid | Sensitive | Sensitive | Sensitive |
| CREC | | | |
| Specimens | Wound | Blood | Urine |
| Ertapenem | Intermediate | Resistant | Resistant |
| Imipenem | Sensitive | Resistant | Intermediate |
| Meropenem | Sensitive | Resistant | Sensitive |
| Cefepime | Sensitive | Resistant | Resistant |
| CRKP | | | |
| Specimens | Bile | Blood | Sputum |
| Ertapenem | Resistant | Resistant | Resistant |
| Imipenem | Resistant | Intermediate | Resistant |
| Meropenem | Resistant | Resistant | Resistant |
| Cefepime | Resistant | Sensitive | Resistant |
| CRPA | | | |
| Specimens | Wound | Blood | Sputum |
| Imipenem | Intermediate | Resistant | Resistant |
| Meropenem | Resistant | Intermediate | Resistant |
| Piperacillin/tazobacter | Sensitive | Sensitive | Intermediate |
| CRAB | | | |
| Specimens | Wound | Blood | Sputum |
| Imipenem | Resistant | Resistant | Resistant |
| Meropenem | Resistant | Resistant | Resistant |
| Levofloxacin | Sensitive | Resistant | Resistant |

## RESULTS

### Effect of 222 nm UVC irradiation on melamine board

In agreement with The Clinical & Laboratory Standards Institute (CLSI) guidelines, 99.9% of bacteria killed is considered the standard of bactericidal efficacy (16). In our experiment, 27.5 mJ/cm$^2$-irradiation achieved a 3-log (99.9%) reduction for MRSA, VRE, and CRKP groups on the melamine board (Fig. 1A, B and D). 18.3 mJ/cm$^2$-irradiation achieved a 3-log (99.9%) reduction for CREC, CRPA, and CRAB groups (Fig. 1C, E and F). For MRSA and CRKP groups, the bacterial counts were reduced to an undetectable level as irradiated with 110.4 mJ/cm$^2$-energy (Fig. 1A and D). The VRE group was reduced to an undetectable level as irradiated with 45.8 mJ/cm$^2$-energy (Fig. 1B). CREC, CRPA, and CRAB groups were reduced to an undetectable level as irradiated with 36.6 mJ/cm$^2$-energy (Fig. 1C, E and F). The bacterial counts of alcohol treatment on six bacterial groups were all undetectable. These results showed that 222 nm UVC irradiation and alcohol treatment have valid bactericidal effects (17) on melamine boards.

### Effect of 222 nm UVC irradiation on silicone rubber

As the radiation energy increased for these six species of bacteria, the number of surviving bacteria decreased. 36.6 mJ/cm$^2$-irradiation achieves 3-log (99.9%) reduction for MRSA, VRE, CREC, and CRAB groups on silicone rubber (Fig. 2A, B, C and F). 45.8 mJ/cm$^2$-irradiation achieved a 3-log (99.9%) reduction for CRKP groups (Fig. 2D). 18.3 mJ/cm$^2$-irradiation achieved a 3-log (99.9%) reduction for CRPA groups (Fig. 2E). MRSA

# Melamine board

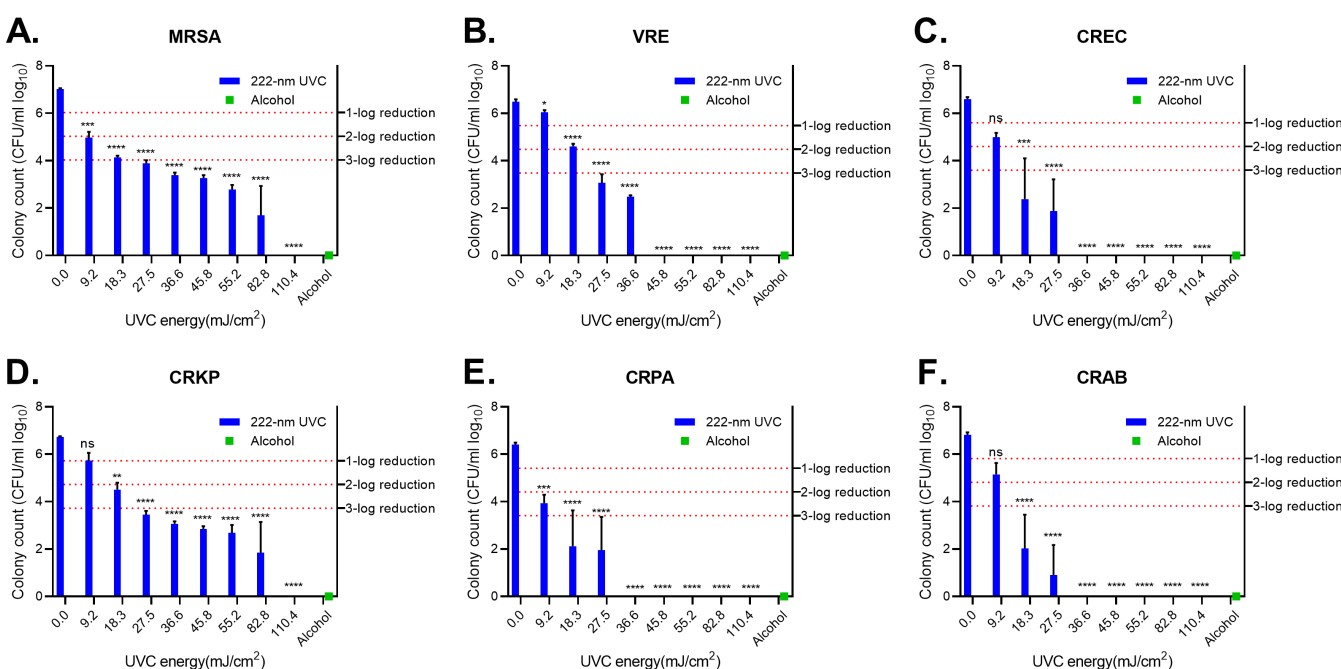

**FIG 1** Disinfection by UVC light on melamine board. (A) MRSA. (B) VRE species. (C) CREC. (D) CRKP. (E) CRPA. (F) CRAB. The effectiveness of Delta U+ Disinfection Device Care 222 in reducing various strains of bacteria was expressed in $\log_{10}$ of colony count. The colony count of alcohol-treatment groups was converted into $\log_{10}$ value and illustrated by green bars. The survival rates adjusted for a 1–3 log reduction: 1-log equals a 90% decrease, 2-log equals a 99% decrease, and 3-log equals a 99.9% decrease from the original concentration. The statistical analyses were performed by using one-way ANOVA. The mean and standard deviation were calculated based on three independent experiments. The error bars indicate the standard variation, and the asterisk denotes the *P*-value (*$P < 0.05$, **$P < 0.01$, ***$P < 0.001$, ****$P < 0.0001$, ns: no significance) compared to the untreated UVC group.

and CRKP groups were reduced to an undetectable level as irradiated with 110.4 mJ/cm$^2$ energy (Fig. 2A and D). VRE, CREC, CRPA, and CRAB groups were reduced to an undetectable level as irradiated with 45.8 mJ/cm$^2$ energy (Fig. 2B, C, E and F). The bacterial counts of alcohol treatment on six bacterial groups were still detectable and had more residual bacteria than 222 nm UVC treatment groups. These results indicate that 222 nm UVC has less effective disinfection ability on silicone rubber than on melamine board, but showed better bactericidal effect than alcohol wipes did.

## Effect of 222 nm UVC irradiation on wood veneer

The 222 nm UVC irradiation treatment on wood veneer had a dose-dependent decreasing tendency but could not sterilize the bacteria completely. In order to examine the efficiency of the sterilized consequence of 222 nm UVC treatment, we prolonged the exposure time. All of the bacteria were completely disinfected after 1 h of 222 nm UVC exposure (Fig. 3). Additional experiments conducted with 2, 4, and 8 h of exposure yielded the same results, with complete eradication of the bacteria. The bacterial counts of alcohol treatment on six bacterial groups were still detectable and had more residual bacteria than 222 nm UVC treatment groups. These results indicate that 222 nm UVC has poor effective disinfection ability on wood veneer than on melamine board, but the disinfection ability could be promoted after prolonging the exposure time.

## DISCUSSION

We confirmed the effectiveness of 222 nm UVC irradiation in disinfecting clinical antibiotic-resistant bacteria on various surfaces. Previous studies have primarily focused on the effectiveness of 222 nm UVC irradiation in deactivating respiratory pathogens

# Silicone rubber

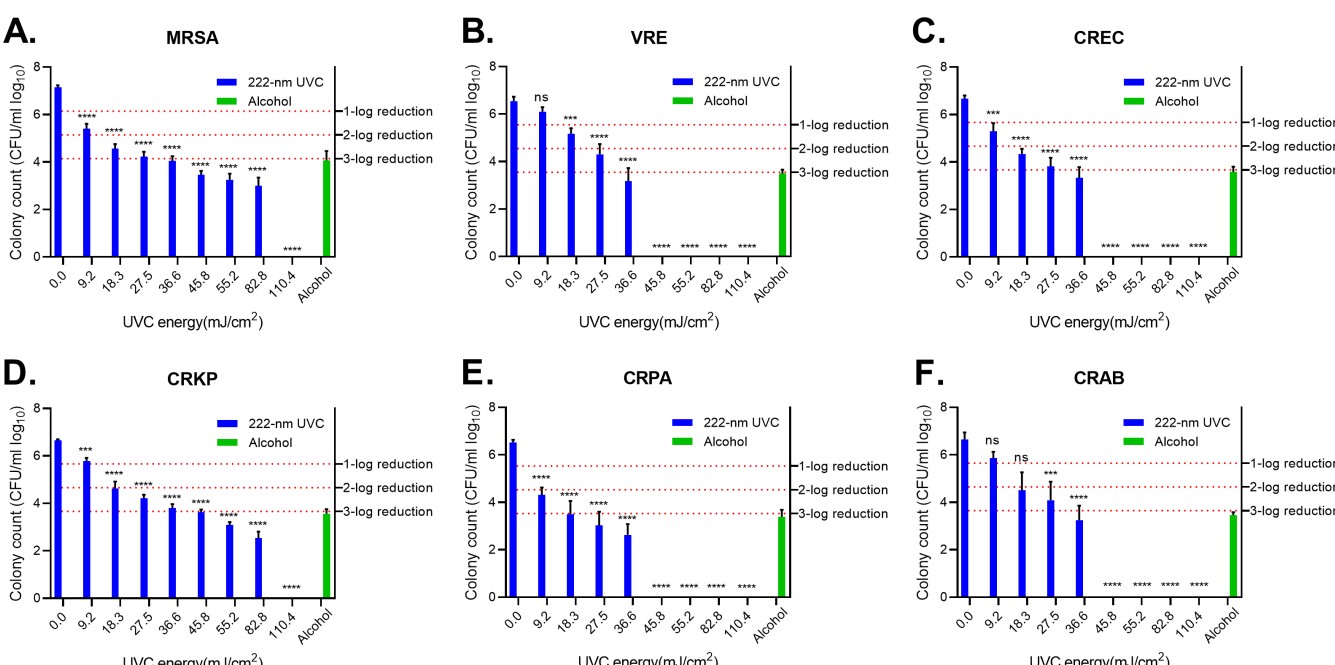

**FIG 2** Disinfection by UVC light on silicone rubber. (A) MRSA. (B) VRE species. (C) CREC. (D) CRKP. (E) CRPA. (F) CRAB. The effectiveness of Delta U+ Disinfection Device Care 222 in reducing various strains of bacteria was expressed in $\log_{10}$ of colony count. The colony count of alcohol-treatment groups was converted into $\log_{10}$ value and illustrated by green bars. The survival rates adjusted for a 1–3 log reduction: 1-log equals a 90% decrease, 2-log equals a 99% decrease, and 3-log equals a 99.9% decrease from the original concentration. The statistical analyses were performed by using one-way ANOVA. The mean and standard deviation were calculated based on three independent experiments. The error bars indicate the standard variation, and the asterisk denotes the *P*-value (*$P < 0.05$, **$P < 0.01$, ***$P < 0.001$, ****$P < 0.0001$, ns: no significance) compared to the untreated UVC group.

that are airborne or transmitted through droplets present in aerosol, including coronaviruses, seasonal and pandemic influenza, and tuberculosis (14, 18). However, the effectiveness of 222 nm UVC irradiation against equipment-mediated bacterial diseases has not been adequately examined. To our knowledge, our study is the first report on the efficacy of 222 nm UVC irradiation against clinical antibiotic-resistant bacteria on commonly used clinical equipment surfaces.

In our investigation, we evaluated the efficacy of 222 nm UVC irradiation for decontaminating various materials and found it to be effective for most surfaces. We focused on three high-touch surfaces and observed that wood veneers with a rough surface retained more bacteria than melamine boards with a smooth surface under the same energy of UVC irradiation. We speculate that the presence of fiber gaps in the wood veneer could potentially diminish the efficacy of UVC irradiation. As a result, elevated levels of irradiation energy were necessary to achieve bacterial sterilization on the wood veneer. Furthermore, the previous research reported that a suspension of MRSA in phosphate-buffered saline (PBS) was exposed to 6 mJ/cm² of 222 nm UVC radiation, resulting in a $10^4$-fold reduction in bacterial count. A further increase in the radiation dose to 12 mJ/cm² led to the number of MRSA being reduced to an undetectable level. In addition, for gram-negative bacteria, a radiation dose of 6–36 mJ/cm² of 222 nm UVC is required to achieve elimination to undetectable levels (19). These low irradiation doses are quite different from our results. The distinction in our study is that we performed the UVC irradiation test on different surfaces in contrast to a previous research, which utilized bacteria suspensions in PBS. We hypothesize that the porosity and absorbency of different surfaces are the main factors influencing the decontaminate ability of 222 nm UVC irradiation. Therefore, rough surfaces require higher doses of 222 nm UVC irradiation energy to achieve effective decontamination.

# Wood veneer

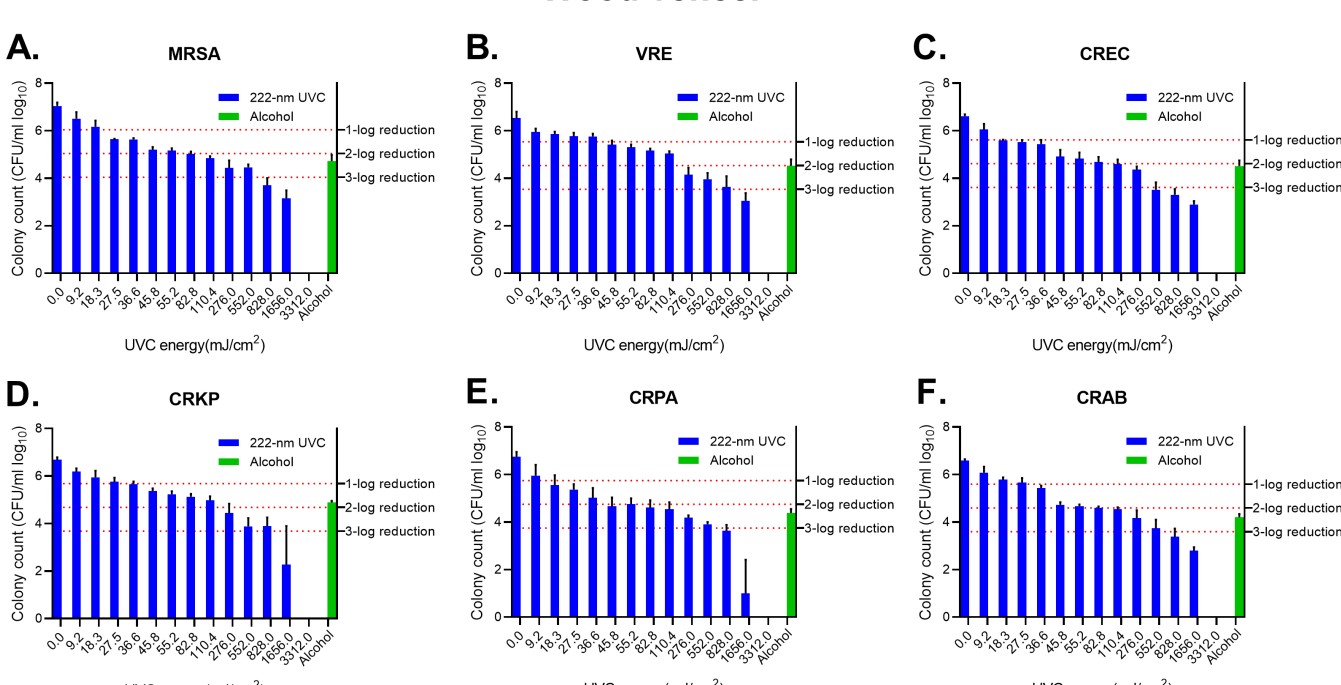

**FIG 3** Disinfection by UVC light on wood veneer. (A) MRSA. (B) VRE species. (C) CREC. (D) CRKP. (E) CRPA. (F) CRAB. The effectiveness of Delta U+ Disinfection Device Care 222 in reducing various strains of bacteria was expressed in $\log_{10}$ of colony count. The colony count of alcohol-treatment groups was converted into $\log_{10}$ values and illustrated by green bars. The survival rates adjusted for a 1–3 log reduction: 1-log equals a 90% decrease, 2-log equals a 99% decrease, and 3-log equals a 99.9% decrease from the original concentration. The mean and standard deviation were calculated based on three independent experiments. The error bars indicate the standard variation.

The bactericidal effects of 222 nm UVC and alcohol wipes on silicone rubber and wood veneer were not ideal. Furthermore, a previous study showed that traditional manual cleaning with alcohol may not always effectively remove pathogens from high-touch surfaces (5). Alcohol, being an organic solvent, may also cause material deterioration and make coatings or painted surfaces brittle for long-term use. Moreover, spraying alcohol on medical instruments can lead to instrument damage due to humidification. To address these limitations of alcohol wipes, we increased the exposure time of 222 nm UVC irradiation (20, 21). FFollowing the application of a higher dose of 222 nm UVC irradiation, the disinfected efficiency could be further improved on these challenging cleaning materials. Taking into account these reasons, 222 nm UVC irradiation is a more suitable and less restricted method than traditional manual cleaning with alcohol.

We observed that different dosages of 222 nm UVC irradiation were required for different bacteria. Specifically, CRKP and MRSA required longer irradiation exposure times compared to other bacteria. We speculate that the structure of the bacteria plays a significant role in their resistance to 222 nm UVC disinfection. Cell capsules of gram-negative bacteria, such as CRKP (22), are mainly composed of polysaccharides that could protect bacteria from toxic compounds and desiccation and allow them to adhere to surfaces (23). Thus, we believe that the membrane structure of bacteria is a critical factor in determining the efficacy of 222 nm UVC disinfection.

In contrast, the thick peptidoglycan (PG) structure of gram-positive bacteria serves a similar function in enhancing their survival in hostile environments (24). Surprisingly, although both MRSA and VRE are gram-positive bacteria, MRSA required more 222 nm UVC energy for inactivation. In accordance with this result, we speculate that two conjectures may explain this difference. Initially, MRSA may have a stronger adhesion ability (25) which necessitates higher energy to eradicate bacteria on surfaces. In addition, a previous study demonstrated that repeated exposures of *S. aureus* to UVC

radiation, combined with multiple growth cycles, resulted in a reduced inactivation effect of UVC on *S. aureus* (26). The MRSA strain examined in our research was sourced from clinical patients exhibiting potential nosocomial infections. We postulate that an extended exposure of MRSA strains in a hospital setting to insufficient UVC sterilization energy could result in the gradual development of increased resistance to UVC radiation over time. Furthermore, with the emergence of the severe acute respiratory syndrome Coronavirus 2 (SARS-CoV-2) epidemic in recent years, there has been a rise in the utilization of household UV germicidal lamps, potentially increasing the likelihood of MRSA exposure to UV radiation. These probable reasons imply that clinical strains are unpredictable and should be considered when designing disinfection protocols.

Biofilm confers bacterial tolerance to environmental threats and facilitates the transfer of antibiotic-resistance genes between bacterial species (27). The formation of biofilm significantly contributes to the development of antibiotic resistance (28). It has been noted that biofilms exhibit reduced sensitivity to UV radiation as they mature, posing challenges for effective disinfection. Current literature predominantly focuses on the effects of UVC at a wavelength of 254 nm on biofilms. Studies suggest that various UVC devices can partially inactivate biofilm-associated cells of *P. aeruginosa*. Notably, UV light emitting diodes (LEDs) emitting at a peak wavelength of 270 nm exhibit the most effective disinfection performance (29). However, further research is warranted to explore the impact of 222 nm UVC on mature biofilms.

In our experiments, we point out that different materials that bacteria attached affect the bactericidal efficacy of 222 nm UVC irradiation. However, it is still unclear whether this method is effective against various types of microorganisms, including molds and bacteria with endospores, which have the function of surviving in harsh environments. Furthermore, the combination of UVC irradiation and detergent wipe may enhance bactericidal efficacy, but further experiments are needed to confirm these assumptions.

In conclusion, our study confirms the sterilized efficiency of 222 nm UVC irradiation against antibiotic-resistant bacteria. However, the efficiency of 222 nm UVC disinfection varies depending on the surface material. Compared to alcohol wipes, 222 nm UVC is a more effective and less restrictive method of disinfection. The required disinfection dose of 222 nm UVC varies for different bacteria, and we speculate that the components of the cell wall and capsule may diminish the efficacy of UVC light. Overall, 222 nm UVC is a valuable disinfection method, but careful attention should be paid to the necessary energy for sterilization on different materials.

## ACKNOWLEDGMENTS

We would like to acknowledge Hui-Yee Yeo, Li-Yun Hsu, Hsiu-Chou Yeh, Shuo-Peng Chou, Yi-Chun Tsai, Yueh-Chiao Chien, and Yu-Shuan Chao at Department of Laboratory Medicine, National Taiwan University Hospital Hsinchu Branch for technical support in our experiment. We would like to express our sincere gratitude to the Intelligent Healthcare Innovation Center (IHIC), National Taiwan University Hospital Hsinchu Branch for their support and assistance in this research.

This work was supported by the National Taiwan University Hospital of the research program, grant no. 112-HCH051.

Conception and design of study: P.-L.S., H.-Y.K., C.-T.T., P.-Y.C. Acquisition of data and data analysis: J.-R.H., T.-W.Y., H.-M.F., K.-H.H. Drafting of manuscript and approval of final version of manuscript: J.-R.H., T.-W.Y., P.-L.S., Y.-I.H.

During the preparation of this work, the authors used ChatGPT May 3 version in order to improve readability and language. After using this service, the authors reviewed and edited the content as needed and take full responsibility for the content of the publication.

## AUTHOR AFFILIATIONS

[1]Department of Laboratory Medicine, National Taiwan University Hospital Hsinchu Branch, Hsinchu, Taiwan

[2]Division of Infectious Disease, Department of Internal Medicine, National Taiwan University Hospital Hsinchu Branch, Hsinchu, Taiwan

[3]Delta Electronics, Inc., Taipei, Taiwan

[4]Department of Otolaryngology, National Taiwan University College of Medicine, Taipei, Taiwan

[5]Intelligent Healthcare Innovation Center, National Taiwan University Hospital Hsinchu Branch, Hsinchu, Taiwan

[6]Department of Pediatrics, National Taiwan University Hospital Hsinchu Branch, Hsinchu, Taiwan

## AUTHOR ORCIDs

Pei-Lan Shao  http://orcid.org/0000-0003-3334-1330

## AUTHOR CONTRIBUTIONS

Jhen-Rong Huang, Data curation, Formal analysis, Investigation, Methodology, Visualization, Writing – original draft | Tsai-Wen Yang, Data curation, Formal analysis, Investigation, Methodology, Software, Visualization, Writing – original draft | Ya-I Hsiao, Project administration, Resources, Supervision, Validation, Writing – review and editing | Hui-Min Fan, Conceptualization, Funding acquisition, Methodology, Project administration | Han-Yueh Kuo, Resources, Supervision, Validation | Kuo-Hsiang Hung, Data curation, Methodology, Resources | Po-Yen Chen, Methodology, Resources | Ching-Ting Tan, Project administration, Supervision, Validation | Pei-Lan Shao, Conceptualization, Project administration, Supervision, Validation, Writing – review and editing

## ADDITIONAL FILES

The following material is available online.

### Open Peer Review

**PEER REVIEW HISTORY (review-history.pdf).** An accounting of the reviewer comments and feedback.

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
