## [Reviewer comments · Microbiology Spectrum]

Microbiology Spectrum

Far-UVC Light (222nm) Efficiently Inactivates Clinically Significant Antibiotic-Resistant Bacteria on Diverse Material Surfaces

Jhen-Rong Huang, Tsai-Wen Yang, Ya-I Hsiao, Hui-Min Fan, Han-Yueh Kuo, Kuo-Hsiang Hung, Po-Yen Chen, Ching-Ting Tan, and Pei-Lan Shao

Corresponding Author(s): Pei-Lan Shao, National Taiwan University Hospital Hsinchu Branch

Review Timeline:

Submission Date:	January 4, 2024
Editorial Decision:	March 15, 2024
Revision Received:	July 1, 2024
Accepted:	August 5, 2024

Editor: Chi-Tai Fang

Reviewer(s): The reviewers have opted to remain anonymous.

Transaction Report:

DOI: <https://doi.org/10.1128/spectrum.04251-23>

Re: Spectrum04251-23 (Far-UVC Light (222nm) Efficiently Inactivates Clinically Significant Antibiotic-Resistant Bacteria on Diverse Material Surfaces)

Dear Dr. Shao:

Thank you for the privilege of reviewing your work. Below you will find my comments, instructions from the Spectrum editorial office, and the reviewer comments.

Revision Guidelines

Sincerely,
Chi-Tai Fang
Editor
Microbiology Spectrum

Reviewer #1 (Comments for the Author):

This work from Huang et al aims at analysing the impact of far-UVC light at 222 nm on bacterial survival on 3 various high-touch surfaces commonly found in healthcare facilities.

It seems that UVC light at 222 nm is powerful to eliminate several critical clinical antibiotic resistant strains, both from G+ and G- bacteria. The more adsorptive the surface is, the more time or intensity of UV light is needed for inactivation.

This work is important for clinical reasons, and could lead to changes in disinfection protocols applied in hospitals worldwide.

The article is very well written. I however propose two major improvements:

- Importantly, in Figures 1, 2 and 3, I thoroughly recommend the authors to use a logarithmic scale for the survival rates in red (as for the colony count in blue), as the visualisation of 99,9% decrease is almost impossible with a classical scale.
- L182, for wood veneer surface, it is surprising that the authors only tested a 8h exposition, whereas in all other cases the longer exposition time is 15 min. I think that the authors should test intermediate times (30 min, 1H, 2H and 4H). These results will be useful to develop future surface sterilization protocols (especially in hospitals).

Minor comments:

- L91 please rephrase, for example "This is because far-UVC light cannot..." or "This is due to low penetration of far-UVC in human cells of skin or eyes, ..."
- L139, please specify what refers to "normal saline"
- L186-189: this very small section correspond to a summary that is also present in the abstract and in the discussion. I think it could be deleted.
- L204: I am not sure that the wood veneers surfaces harbour more bacteria. Perhaps they are protected by the adsorption, as proposed by the authors?
- L233-243: as stated by the authors themselves, the VRE are G+ bacteria as MSRA. Thus, I think the speculation that the thick peptidoglycan of G+ could play a role in the MRSA resistance (l.237-238) is not valid.
- L248: I do not understand why MRSA clinical isolates could have been exposed to UVC radiation?? Please explain
- In the Figures 1, 2 and 3, please replace Alcohol by Alcohol

Reviewer #2 (Comments for the Author):

This study is aimed to evaluate the efficacy of 222 nm- UVC light to inactivate MDR bacteria on diverse nosocomial surfaces. The work overall is sound. The writing style needs to be improved. Additional concerns are as follow:

- I consider that a big limitation of this study is to make general conclusions regarding the efficacy of UVC radiation on different clinically-relevant bacterial species by only testing one strain of each species. I strongly suggest to address this issue.
- L50. This result should include the time of exposition to UVC light.
- L66. Could the authors better explain the concept: "the widespread use of antibiotics which led to the evolution of bacteria"?
- L123-128. Please, include more information about the bacterial isolates, such as from what kind of infections were isolated, detailed antibiotic resistant pattern, ethical protocol number allowing to use the isolates for scientific research.
- L131. Why the growing conditions include 5% CO₂?
- L138-144. The experimental conditions should be better explained. What was the surface area used to apply 1.5×10^7 CFU? Were the surfaces previously sterilized? How? In which volume was the bacteria collected from the surface after UVC treatment? How many replicate determinations were performed in each independent assay? How many independent assays were performed?
- Please, add to the Materials and Methods section the statistical analysis performed. Legends of Figs. 1-3 indicate that paired t-test were used, however it will be more appropriate to use ANOVA.
- L154. Figure 2B should be changed by Figure 1B.
- L156. Figure 2 should say Figure 1.
- L186-189 and Table 2. Authors are repeating results showed in Figs 1-3. Please remove both this text and Table 2
- What would be the efficacy of 222 nm-UVC radiation on mature biofilms? Please discuss.
- Legends of Figs. 1-3. Please add how many independent experiments were performed to calculate the mean and SD.

This work from Huang et al aims at analysing the impact of far-UVC light at 222 nm on bacterial survival on 3 various high-touch surfaces commonly found in healthcare facilities.

It seems that UVC light at 222 nm is powerful to eliminate several critical clinical antibiotic resistant strains, both from G+ and G- bacteria. The more adsorptive the surface is, the more time or intensity of UV light is needed for inactivation.

This work is important for clinical reasons, and could lead to changes in disinfection protocols applied in hospitals worldwide.

The article is very well written. I however propose two major improvements:

- Importantly, in Figures 1, 2 and 3, I thoroughly recommend the authors to use a logarithmic scale for the survival rates in red (as for the colony count in blue), as the visualisation of 99,9% decrease is almost impossible with a classical scale.
- L182, for wood veneer surface, it is surprising that the authors only tested a 8h exposition, whereas in all other cases the longer exposition time is 15 min. I think that the authors should test intermediate times (30 min, 1H, 2H and 4H). These results will be useful to develop future surface sterilization protocols (especially in hospitals).

Minor comments:

- L91 please rephrase, for example “This is because far-UVC light cannot...” or “This is due to low penetration of far-UVC in human cells of skin or eyes, ...”
- L139, please specify what refers to “normal saline”
- L186-189: this very small section correspond to a summary that is also present in the abstract and in the discussion. I think it could be deleted.
- L204: I am not sure that the wood veneers surfaces harbour more bacteria. Perhaps they are protected by the adsorption, as proposed by the authors?
- L233-243: as stated by the authors themselves, the VRE are G+ bacteria as MRSA. Thus, I think the speculation that the thick peptidoglycan of G+ could play a role in the MRSA resistance (l.237-238) is not valid.
- L248: I do not understand why MRSA clinical isolates could have been exposed to UVC radiation?? Please explain
- In the Figures 1, 2 and 3, please replace Alchohol by Alcohol

Dear Dr. Fang,

We are pleased to submit the revised draft of our manuscript, “Far-UVC Light (222nm) Efficiently Inactivates Clinically Significant Antibiotic-Resistant Bacteria on Diverse Material Surfaces” (#Spectrum04251-23), to *Microbiology Spectrum*. We appreciate the time and effort dedicated by the editorial staff and reviewers. The comments provided were valuable and helped us refine our paper. As such, we have made several revisions to the manuscript based on the suggestions given. Changes to the manuscript are highlighted.

Below are our point-by-point responses to the reviewers’ comments.

Response to Reviewer 1

Thank you for your insightful comments and suggestions. Please find the answers to each of your questions below.

1. Importantly, in Figures 1, 2 and 3, I thoroughly recommend the authors to use a logarithmic scale for the survival rates in red (as for the colony count in blue), as the visualisation of 99,9% decrease is almost impossible with a classical scale.

Response: Thank you for your close reading of our paper. The survival rates adjusted for a 1-3 log reduction: 1-log equals a 90% decrease, 2-log equals a 99% decrease, and 3-log equals a 99.9% decrease from the original concentration. (L311-313, 327-329, 342-344)

2. L182, for wood veneer surface, it is surprising that the authors only tested a 8h exposition, whereas in all other cases the longer exposition time is 15 min. I think that the authors should test intermediate times (30 min, 1H, 2H and 4H). These results will be useful to develop future surface sterilization protocols (especially in hospitals).

Response: We agree with your suggestion and increase the time in between. Surprisingly, we found out that after 1 hour of 222–nm UVC treatment, all six species of bacteria were killed. The results were added in Figure 3. Additional experiments

conducted with 2 hours, 4 hours, and 8 hours of exposure yielded the same results, with complete eradication of the bacteria. (L194-197)

3. **L91 please rephrase, for example "This is because far-UVC light cannot..." or "This is due to low penetration of far-UVC in human cells of skin or eyes, ..."**

Response: We apologize for this error. The sentence has been modified based on your suggestions. (L93-95)

4. **L139, please specify what refers to "normal saline".**

Response: Normal saline refers to 0.9% sodium chloride solution. The text has been modified to 0.45% sodium chloride solution. 0.45% sodium chloride solution were used in our clinical operations to determine bacterial susceptibility. (L143-144)

5. **L186-189: this very small section correspond to a summary that is also present in the abstract and in the discussion. I think it could be deleted.**

Response: This paragraph has been deleted as suggested.

6. **L204: I am not sure that the wood veneers surfaces harbour more bacteria. Perhaps they are protected by the adsorption, as proposed by the authors?**

Response: Wood veneers retained more bacteria than melamine boards under the same energy of UVC irradiation. We speculate that the presence of fiber gaps in the wood veneer could potentially diminish the efficacy of UVC irradiation. (L214-220)

7. **L233-243: as stated by the authors themselves, the VRE are G+ bacteria as MRSA. Thus, I think the speculation that the thick peptidoglycan of G+ could play a role in the MRSA resistance (l.237-238) is not valid.**

Response: Both MRSA and VRE are Gram-positive bacteria, but MRSA requires more 222-nm UVC energy for inactivation. Based on this result, we speculate that two conjectures may explain this difference. Firstly, MRSA may have a stronger adhesion ability which necessitates higher energy to eradicate bacteria on surfaces. Secondly, a previous study demonstrated that repeated exposures of *Staphylococcus*

aureus to UVC radiation combined with multiple growth cycles, resulted in a reduced inactivation effect of UVC on *S. aureus*. (L255-263)

8. L248: I do not understand why MRSA clinical isolates could have been exposed to UVC radiation?? Please explain

Response: The MRSA strain examined in our research was sourced from clinical patients exhibiting potential nosocomial infections. The regulations of our hospital stipulate that MRSA patients do not require isolation unless they are also resistant to sulfamethoxazole-trimethoprim (SXT). Therefore, we hypothesize that MRSA strains exposed to insufficient UVC sterilization energy over extended periods in the hospital environment may gradually develop increased resistance to UVC radiation. Furthermore, with the emergence of the SARS-CoV-2 epidemic in recent years, there has been a rise in the utilization of household UV germicidal lamps, potentially increasing the likelihood of MRSA exposure to UV radiation. (L263-270)

9. In the Figures 1, 2 and 3, please replace Alchohol by Alcohol

Response: We apologize for this error. Corrected misspelled words.

Response to Reviewer 2

We appreciate your careful review of our paper. Our answers are as follows.

- 1. I consider that a big limitation of this study is to make general conclusions regarding the efficacy of UVC radiation on different clinically-relevant bacterial species by only testing one strain of each species. I strongly suggest to address this issue.**

Response: Thank you for your suggestion. We added 2 more strains for each species (L130-132). After adding two new strains, the results are slightly different from the previous data. (L167-170, L177-184, L194-201)

- 2. L50. This result should include the time of exposition to UVC light.**

Response: Thank you for your suggestion. We added the time of exposition to UVC light. The prolonged exposure time added in materials and methods. (L52-53, L149-151)

- 3. L66. Could the authors better explain the concept: "the widespread use of antibiotics which led to the evolution of bacteria"?**

Response: Thank you for your close reading of our paper. Previous studies mentioned that the widespread use of antibiotics has led to the evolution of bacteria to evade antibiotic attacks. (Munita JM, Arias CA. Microbiol Spectr. 2016 Apr; PMID: PMC4888801.)(L65-68)

The sentence has been modified based on your suggestions.

- 4. L123-128. Please, include more information about the bacterial isolates, such as from what kind of infections were isolated, detailed antibiotic resistant pattern, ethical protocol number allowing to use the isolates for scientific research.**

Response: The bacterial strains were isolated from different specimens. The details were recorded in Table 2.

The ethical protocol number is NTUH-REC No.: 202301215W. The document was uploaded in file type of Miscellaneous File not for Publication. (L132-133)

5. L131. Why the growing conditions include 5% CO₂?

Response: Our clinical specimens are cultured in a 5% CO₂ incubator; therefore, all experiments are also conducted in a 5% CO₂ incubator.

6. L138-144. The experimental conditions should be better explained. What was the surface area used to apply 1.5x10⁷ CFU? Were the surfaces previously sterilized? How? In which volume was the bacteria collected from the surface after UVC treatment? How many replicate determinations were performed in each independent assay? How many independent assays were performed?

Response:

1. The surface area is 2.5 cm². (L151-152)

2. The materials were all pretreated with 75% alcohol and 256nm-UV light for 30 minutes. (L146-147)

3. eSwabs were used to collect the surface area, by rolling it back and forth 10 times and inoculating it onto a TSA agar plate. (L151-153)

4 and 5. The replicate determinations were performed two times in each independent assay, and independent assays were performed three times for each strain. (L155-158)

7. Please, add to the Materials and Methods section the statistical analysis performed. Legends of Figs. 1-3 indicate that paired t-test were used, however it will be more appropriate to use ANOVA.

Response: Thank you for your close reading of our paper. We have modified the statistical method to one-way ANOVA. (L157-158)

8. L154. Figure 2B should be changed by Figure 1B.

Response: We apologize for this error. We have corrected the text.

9. L156. Figure 2 should say Figure 1.

Response: We apologize for this error. We have corrected the text.

**10. L186-189 and Table 2. Authors are repeating results showed in Figs 1-3.
Please remove both this text and Table 2**

Response: This paragraph has been deleted as suggested.

**11. What would be the efficacy of 222 nm-UVC radiation on mature biofilms?
Please discuss.**

Response: Thank you for your suggestion. We added a discussion on the effect of 222 nm-UVC on mature biofilms. (L274-283)

**12. Legends of Figs. 1-3. Please add how many independent experiments were
performed to calculate the mean and SD.**

Response: Thank you for your suggestion. We revised the figure legend.

Additionally, we have thoroughly proofread the manuscript to eliminate any remaining grammatical and spelling errors.

We look forward to your response regarding our submission. Please do not hesitate to contact us if there are any further questions or comments.

Sincerely,

Pei-Lan Shao

Corresponding author

Department of Laboratory Medicine, National Taiwan University Hospital Hsinchu Branch.

Re: Spectrum04251-23R1 (Far-UVC Light (222nm) Efficiently Inactivates Clinically Significant Antibiotic-Resistant Bacteria on Diverse Material Surfaces)

Dear Dr. Pei-Lan Shao:

Your manuscript has been accepted, and I am forwarding it to the ASM production staff for publication. Your paper will first be checked to make sure all elements meet the technical requirements. ASM staff will contact you if anything needs to be revised before copyediting and production can begin. Otherwise, you will be notified when your proofs are ready to be viewed.

Sincerely,
Chi-Tai Fang, MD, PhD
Editor
Microbiology Spectrum

Reviewer #1 (Comments for the Author):

I thank the authors for taking in consideration all of my comments.